# ONE FOR MANY: AN INSTAGRAM INSPIRED BLACK-BOX ADVERSARIAL ATTACK

## ABSTRACT

We propose a nested evolutionary algorithm able to craft multi-network (decision-based) black-box adversarial attacks based on Instagram inspired image filters. Due to the multi-network training, the system reaches a high transferability rate of attacks and, due to the composition of image filters, it is able to bypass standard detection mechanisms. Moreover, this kind of attack is semantically robust: our filter composition cannot be distinguished from any other filter composition used extensively every day to enhance images; this raises new security issues and challenges for real-world systems. Experimental results demonstrate that the method is also effective against ensemble-adversarially trained models and it has a low cost in terms of queries to the victim model.

## 1 INTRODUCTION

It is well known that deep learning models are susceptible to adversarial attacks and many recent researches in the field have been devoted to produce ever more reliable and effective attacks. Attack reliability is strictly connected to its applicability in real-world scenarios and to its ability to bypass potential defense mechanisms; this is why the hard-label black-box setting (also called decision-based attack) and the transferability property of white box attacks gained increasing attention. Several techniques have been proposed to increase the transferability of both black-box and white-box attacks (Cheng et al. (2020); Wu et al. (2020); Brendel et al. (2018) , among the most popular). One of the most commonly adopted technique is to craft the attacks by using an ensemble of multiple models as proposed by Liu et al. (2017).

Besides the categorization in white-box and black-box methods, attacks can be classified as *restricted* or *unrestricted*, considering the amount of modifications they apply to the images in order to fool the systems.
**Restricted attacks:** they generally use a $L_p$-norm distance to bound the modifications. The attacks are crafted with the aim of minimizing the differences between the original image and the adversarial one, even if it means having visible (more or less) artifacts. In Figures 1b and 1d, the attacks produced by Dong et al. (2019) and Wu et al. (2020), two of the most recent state-of-the-art adversarial methods, are shown. In both the cases the generated artifacts are clearly visible.
**Unrestricted attacks:** they employ large and visible perturbations while keeping the images realistic, natural looking and non-suspicious. The idea is to obtain images that can admit great differences from the original one but, beyond a direct comparison with the original one they cannot be distinguished from any other real (maybe filtered) image. In Figures 1f and 1h, the attacks produced by ACE-Ins (Zhao et al., 2020) and Colorfool (Shahin Shamsabadi et al., 2020) are shown. The differences between the original image and the adversarial one are evident but, if the modifications are good enough, looking just at the adversarial one we might not be able to say that it is the attacking image.

Another aspect that has to be taken into account for real-world attacks is the amount of queries to the victim model that are necessary to craft effective attacks. All the systems proposed in literature need a huge amount of queries and, also in the case of systems built to work with limited access to the victim model, several thousands of queries are needed to produce reliable attacks.

In this paper we propose a system to craft reliable, effective and highly transferable attacks by composing image filters. In previous papers (for ex. Destylization) has been shown that Instagram

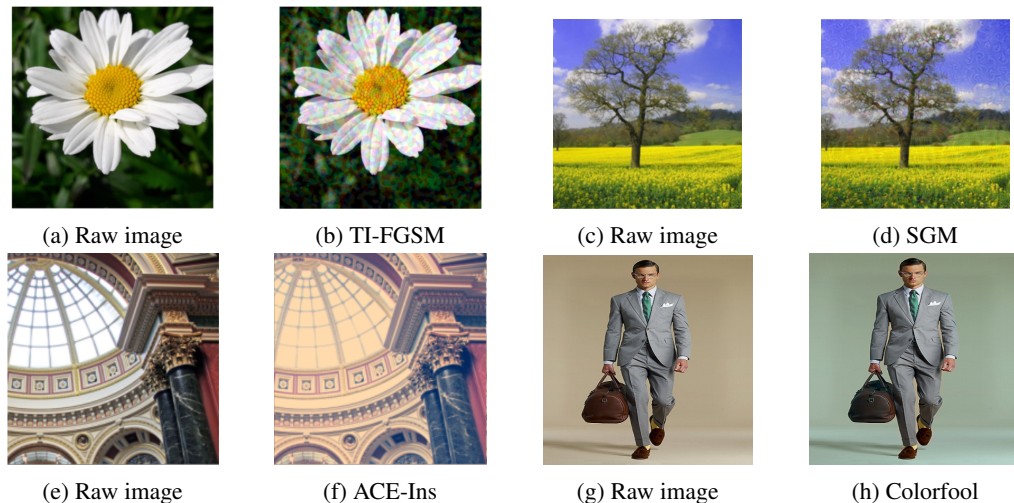

| (a) Raw image | (b) TI-FGSM | (c) Raw image | (d) SGM |

| (e) Raw image | (f) ACE-Ins | (g) Raw image | (h) Colorfool |

Figure 1: Examples of adversarial samples generated by TI-FGSM (Dong et al., 2019), SGM (Wu et al., 2020), ACE-Ins (Zhao et al., 2020), and Colorfool (Shahin Shamsabadi et al., 2020).

inspired filters can have adversarial, but limited effects, when applied singularly. Hence, we decided to study the effects of filter composition with the aim to increase the adversarial effectiveness. Moreover, in the usual photo editing process people use more than one filter in order to obtain the desired effect. We just wanted to simulate a more realistic scenario.

This kind of attack offers multiple benefits: *(i)* it is naturally robust to detecting methods able to find noise, injected patterns and irregularities in the high frequencies image (Moosavi-Dezfooli et al., 2018; Liao et al., 2018). *(ii)* it is naturally robust to masking-gradients (and their numerical estimate) defending methods. *(iii)* it produces natural looking artifacts-free images. *(iv)* it produces adversarial attacks that cannot be distinguished from any other filtered images produced extensively every day, especially on social media platforms.

The system works in the pure hard-label black box setting and implements a multi-network approach to increase the attack transferability. It can reach, with artifact-free images, a transferability rate higher than 70% in the case of unsecured networks and, even more interestingly, higher than 60% in the case of adversarially trained network.

Moreover, the system requires a very low number of queries to find an attack, around 640 in the average case, and it can be limited by construction adjusting the algorithm parameters. In the experiments presented here the maximum number of allowed queries is 1610.

Our contribution can be summarized as follows: *(i)* We propose the AGV-multinetwork attack, a system able to craft powerful adversarial attacks capable of attacking even secured adversarially trained models by composing Instagram inspired image filters; *(ii)* we empirically demonstrate the high transferability of these attacks and we compare our results with other state-of-the-art systems; *(iii)* we empirically demonstrate the efficiency of our system in terms of queries to the victim model and we compare our results with other state-of-the-art systems.

## 2 BACKGROUND

### 2.1 ADVERSARIAL MACHINE LEARNING

Given an input image $x \in X \subset \mathbb{R}^d$ and its corresponding label $y$, let $F$ be a neural network classifier that (correctly) predicts the class label for the input image $x : F(x) = y$. An adversarial attack attempts to modify the input image $x$ by adding a perturbation $\delta$ into an adversarial image $x^* = x + \delta$ such that the classifier is mislead into making a wrong prediction, i.e. $F(x^*) \neq F(x)$.

If we consider the type of the applied perturbation $\delta$, the attacks can be classified as restricted or unrestricted. In the restricted case, the modifications applied to the original image are usually small

and bounded by a $L_p$-norm distance measure, forcing the adversarial image $x^*$ to be as close as possible to the original one. On the contrary, unrestricted attacks use large perturbations without $L_p$-bounded constraints that manipulate the image in order to create photo-realistic adversarial examples. In this case the objective is not to limit the modifications on pixels but limit the human perception that a modification has been applied (Shahin Shamsabadi et al., 2020; Zhao et al.; Wang et al., 2021).

## 2.2 IMAGE FILTERS

We implemented ten of the most popular Instagram filters using Python3 and the Pillow, OpenCV and Numpy libraries: *Clarendon, Juno, Reyes, Gingham, Lark, Hudson, Slumber, Stinson, Rise,* and *Perpetua*. Each filter has distinct characteristics and effects given by different level of contrast, saturation, brightness, shadows, etc. For instance, *Clarendon* brightens and highlights a photo, *Juno* adds saturation and warmth making the colors more intense, *Rise* gives a warm glow by mixing a radial gradient with a light sepia tone, while *Hudson* bumps up the blues giving a more colder feel. Examples of single filter applications are shown in Figure 6 in Appendix A. .

Each filter is parameterized by two parameters that have to be optimized by the algorithm: *intensity* $\alpha$ and *strength* $s$. The role of the parameter $\alpha$ is to alter the intensity of each basic component inside each filter implementation such as brightness, contrast, saturation, edge enhancement, gamma correction and many more. The parameter $s$ is used to control the strength of the filter application and it is defined as the convex interpolation between the original image $x$ and the transformed image $x^*$:

$$strength(x, x^*, s) = (1.0 - s) \cdot x + s \cdot x^* \tag{1}$$

thus, if $s = 0$ the output image of the filter is the original image, while with $s = 1$ the filter returns the modified image $x^*$.

# 3 APPROACH AND ALGORITHM

## 3.1 PROBLEM DEFINITION

Given a set $S = \{f_1, f_2, \ldots, f_m\}$ of Instagram inspired image filters as described in Section 2.2, and a clean image $x$, we want to find a sequence of $n$ parametrized filters $\{f_{k_1}(\alpha_1, s_1), \ldots, f_{k_n}(\alpha_n, s_n)\}$ able to produce an adversarial attack against a classifier model $F$ starting from the image $x$, that is

$$\begin{cases} F(x) \neq F(x^*) \\ x^* = f_{k_n}(\alpha_n, s_n)(\ldots f_{k_2}(\alpha_2, s_2)(f_{k_1}(\alpha_1, s_1)(x))) \end{cases} \tag{2}$$

## 3.2 APPROACH

The algorithm used to optimize the sequence of filters and their parameters is inspired by Baia et al. (2021). In our proposal the universal approach proposed by Baia et al. has been transformed into a *multi-network per-image* approach, where the attack is crafted and optimized for just one image with respect to more target models.

Since the algorithm uses an evolutionary approach, the fitness function used to guide the search can have any (also non differentiable) form, and we decide to use it to implement a sort of generalization ability induced by attacking more models simultaneously. This is inspired by Liu et al. (2017) where the authors suggested that attacking an ensemble of multiple networks simultaneously can generate much stronger adversarial examples. Instead of attacking an ensemble network we decided to exploit the optimization ability of the evolutionary approach and try to simultaneously attack all the (or at least $k$) reference networks. Therefore, our goal is to find *one* adversarial perturbation per-image that can fool *many* deep learning models. This objective is much harder to achieve than attacking a single model but it guarantees complete (100%) transferability of the attack towards the reference networks.

The proposed multi-network approach has the objective to obtain adversarial images with better transferability avoiding the natural overfitting trap we can fall into when the attacks are crafted by using a unique reference model.

### 3.3 ALGORITHM

The optimization method consists of two evolutionary nested algorithms: the outer algorithm, using a GA approach, in charge of finding the sequence of filters to use, and the inner algorithm, based on ES, used to choose the filter parameter values. The general structure of the multi-network algorithm is shown in Algorithm 1. Given a set $S = \{f_1, f_2, f_m\}$ of image filters, the outer algorithm *genotype* (with length $n$) is encoded as a list of $n$ integers $k_1, \ldots, k_n \in \{1, \ldots, m\}$ representing the corresponding filters in $S$, while the inner algorithm *genotype* is represented by a list containing the pairs of parameters used for each selected filter $((\alpha_1, s_1), \ldots, (\alpha_n, s_n))$. The associated *phenotype* is sequence of parametrized filters able to generate the adversarial examples by applying the selected sequence of filters, with their corresponding optimized parameters, as described in Eq.2.

#### 3.3.1 OUTER ALGORITHM

The outer optimization step is performed by a genetic algorithm: a population of $N$ candidate perturbations is iteratively evolved towards better solutions. In order to breed a new generation, population members are randomly selected and the crossover and mutation operations are performed. The quality of the candidates is evaluated based on their fitness values and, at the end of each iteration, the $N$ best individuals are chosen for the next generation.

**Initial population:** it is generated by randomly selecting $l$ filters from the set $S$ of available filters and their parameters are initialized with default values equal to 1.

**Crossover:** a standard one-point crossover is used to generate new off-springs from randomly selected members. Each child is guaranteed to inherit some genetic information from both parents, including the optimized parameters. For example, given two parent elements $P_1 = (f_1'(\alpha_1', s_1'), \ldots, f_n'(\alpha_n', s_n'))$ and $P_2 = (f_1''(\alpha_1'', s_1''), \ldots, f_n''(\alpha_n'', s_n''))$ and crossover index $i = 2$, we obtain the child element $(f_1'(\alpha_1', s_1'), f_2'(\alpha_2', s_2'), f_3''(\alpha_3'', s_3''), \ldots, f_n''(\alpha_n'', s_n''))$.

**Mutation:** it is applied by substituting a filter with another one based on a mutation probability. The substituent filter is initialized with random parameter values. For example considering the element $P = (f_1(\alpha_1, s_1), f_2(\alpha_2, s_2), f_3(\alpha_3, s_3) \ldots, f_n(\alpha_n, s_n))$ and supposing that filters $f_1$ and $f_3$ has been chosen to mutate with $g_1$ and $g_2$, the new generated element is $P^* = (g_1(\alpha_1^*, s_1^*), f_2(\alpha_2, s_2), g_2(\alpha_2^*, s_2^*) \ldots, f_n(\alpha_n, s_n))$, where $\alpha_i^*, s_i^*$ are randomly extracted from the parameter domains.

**Selection:** at the end of each iteration, we choose the N best individuals from the set of 2N candidates (parents and offsprings) according to their fitness values. This process is repeated until the algorithm exhausts the allowed number of generations.

#### 3.3.2 INNER ALGORITHM

This algorithm is devoted to the parameter optimization and is achieved by using $(1, \lambda)$ evolution strategy with $\lambda = 5$. ES iteratively updates a search distribution by following the natural gradient towards higher expected fitness. In our case, for each list of parameters we compute a batch of $\lambda$ samples by perturbing the original individual. A gradient towards a better solution is estimated using the fitness values of the $\lambda$ samples. This gradient is then used to update the original individual. This process is repeated until the algorithm reaches the allowed number of iterations.

### 3.4 EVALUATION

We define the fitness function based on the attack success rate (*ASR*) since the objective of the algorithm is to fool image classification systems.

In the case of multi-networks attacks we average the attack success rate values computed on multiple networks simultaneously. Specifically, given $K$ $F_i$ neural networks models, a $I$ legitimate image, and the corresponding perturbed image $I^*$ obtained by applying the sequence of parametrized filters we want to evaluate, we defined the multi-network fitness function as:

$$\mathcal{F}_\mathcal{K}(I, I^*) = \frac{1}{K} \sum_{k=1}^{K} ASR_k(I, I^*), \tag{3}$$

where $ASR_k$ represents the attack success rate calculated by querying the $k$-th target model: it returns 1 if $F_k(I) \neq F_k(I^*)$, 0 otherwise with respect to the Top-1 prediction.

### 3.4.1 ALLOWED QUERIES

The algorithm works with limited accesses to the victim model. In our algorithm the victim model is called every time we have to compute the fitness function. Besides the explicit calls to the evaluation function made in the initialization step and in the last step of each outer iteration, we have to consider the $E_{in} \times N_{in}$ calls made by the inner optimization phase (filters' parameters optimization) for each element in the outer population. The maximum number of allowed queries can be easily computed as:

$$Q_{MAX} = N_{out} + E_{out} \times (N_{out} \times E_{in} \times N_{in} + N_{out}) \tag{4}$$

The drawback of using an evolutionary approach is highly mitigated by the needed reduced number of generations and population size.

---

**Algorithm 1:** AGV-multi-network algorithm

---

**Input**: Image $I$, $K$ classification models, outer population size $N_{out}$, inner population size $N_{in}$, number of outer generations $E_{out}$, number of inner generations $E_{in}$, $obj$ the number of target models we want to attack.

Initialize population $P$ of $N_{out}$ individuals;

**foreach** $p \in P$ **do**
    | Evaluate the fitness of $p$ by $\mathcal{F}_\mathcal{K}$ ;
**end**

e=0 ;

**while** *($e < E_{out}$ **and** $\mathcal{F}_\mathcal{K} \cdot K < obj$)* **do**
    Offsprings = $\{\emptyset\}$ ;
    **for** $i = 1$ **to** $N_{out}$ **do**
        Select randomly two parents $p_1$, $p_2$ from $P$ ;
        $y_i$ = crossover$(p_1, p_2)$;
        $\overline{y}_i$ = mutation$(y_i)$ ;
        $n_i$ = ES_optimizer$(\overline{y}_i, E_{in}, N_{in})$ ;
        Offsprings $\leftarrow$ ( $n_i$) ;
    **end**
    **foreach** $o \in$ Offsprings **do**
        | Evaluate the fitness of $o$ by $\mathcal{F}_\mathcal{K}$;
    **end**
    $P$ = selection$(P,$ Offsprings$)$ ;
    $e = e + 1$ ;
**end**

**return**: best filter sequence for image $I$;

---

## 4 EXPERIMENTS

### 4.1 EXPERIMENTAL SETUP

**Networks**: To validate the proposed method we employed five state-of-the-art neural networks for image classification as reference models to find the adversarial filter sequences, namely ResNet50, VGG19, DenseNet201, MobileNet and NasNetMobile. To evaluate the effectiveness of the transferability we use as hold-out models two normally trained networks, i.e. InceptionV3 and InceptionV4, and one adversarially trained network, i.e InceptionResNetV2-ens-adv (Tramèr et al., 2018).

Different architectures were chosen in order to explore the generalization capability of the algorithm given that these models are fundamental components of other deep learning models used for object detection and image segmentation. Besides, we decided to use MobileNet and NasNetMobile because they are architectures often used in ML-powered mobile apps thanks to their low-latency and lightweight nature which makes them an interesting case study. These models were trained on the large visual ImageNet training dataset (Deng et al., 2009) and they are publicly available.

**Multi-network setup**: In the multi-network configuration we simultaneously attack multiple deep learning models. More specifically, we study three multi-network setups:

- **5-NET**: includes MobileNet, ResNet50, VGG19, DenseNet201, NasNetMobile
- **4-NET**: includes MobileNet, ResNet50, VGG19, DenseNet201
- **3-NET**: includes MobileNet, ResNet50, VGG19

**Dataset**: We performed the experiments on a subset of 1000 images from the ImageNet validation dataset. The images were preprocessed according to the requirements of each pre-trained neural network.

**Implementations details.** The hyperparameters of the outer algorithm were set as follows: $N_{out} = 10$, $E_{out} = 10$ and mutation probability $\rho = 0.5$; The hyperparameters of the inner algorithm were set as follows: $N_{in} = 5$, $E_{in} = 3$, initial learning rate = 0.1 and decay rate = 0.75. These values are the result of a preliminary experimental phase conducted to find a good trade-off between performance and computational time.

## 4.2 QUERY EFFICIENCY

Query efficiency is considered a key characteristic for generating realistic attacks. As highlighted by Ilyas et al. (2018) a limit on the number of queries can result from a time limit or a budget limit, when querying the system costs. Considering the settings used, our system allows a maximum of 1610 queries to the victim model for each network used in the multi-network system. Direct comparisons with other methods are in general difficult due to the differences of the approaches, especially in the case of $L_p$-bounded modifications. We report the results obtained on the ImageNet dataset by very recent systems that can be considered query-efficient methods working in the hard-label black-box settings: Ilyas et al. (2018), that is based on an evolutionary strategy, reports that effective attacks can be crafted with 270k queries to obtain the 90% of accuracy; in Li et al. (2020) the authors report that their best system can produce attacks with a good quality in the 76% of cases with 10k queries or in the 98% of cases with 20k queries. In Cheng et al. (2020) the authors report that they can reach the 50% of accuracy with 30k queries, but they need 160k queries to reach the 90%. It is clear that, comparing these results with our maximum (that varies from 1610 for single network training to 8050 for 5-networks training), AGV-multinetwork is more efficient in all the cases.

## 4.3 ATTACKS AND TRANSFERABILITY

In this section, we analyze the multi-network scenario. According to (Liu et al., 2017), if an adversarial instance remains adversarial for multiple networks then it is more likely to transfer to other unseen networks as well. Therefore, we leverage the capabilities of the AGV single-model by (Baia et al., 2021) algorithm to attack multiple neural networks simultaneously in order to improve the adversarial robustness and generate adversarial examples with high transferability rates.

We examine the three different multi-network configurations 5-NET, 4-NET and 3-NET described before with filter sequences of length 3, 4, and 5. In this case, we consider a combination of filters to be adversarial if it is capable of misleading all models included in the multi-network setup with respect to the Top-1 predicted class. This is the strongest requirement because we ask the system to find an adversarial filter composition that is able to fool all the networks in the multi-network configuration. This is different from attacking an ensemble network: we guarantee that these images (when they are found) surely will fool all the networks used, while this is not guaranteed by the ensemble training. For example Xie et al. (2019) showed in their experiments for DI$^2$-FGSM and M-DI$^2$-FGSM that the attacks trained with the ensemble reach the 90% ASR just in few cases. Results are reported in Table 1, where the Single-network columns can be considered the baseline results obtained by running the algorithm proposed by Baia et al. (2021) on a single image. As expected, attacking multiple networks is a more difficult task than attacking single models. However, since the multi-network adversarial examples are guaranteed to work across all built-in models this reduces the need to run the algorithm for each single network individually. Unsurprisingly, there is a correlation between the number of filters and the ASR: longer sequences produce more visible and large modifications that enables to guide an image towards an adversarial sample more easily.

We also computed the transferability using three test models, namely InceptionV3, InceptionV4, and the adversarially trained InceptionResnetV2-ens-adv, for both multiple and single model attacks. The results are shown in Table 2. In this case, we can notice that the multi-network based attacks achieve higher transferability rates compared to the single-model setup.

Moreover, our attacks can also mislead InceptionResNetV2-ens-adv on more than 60% of the adversarial samples. This shows that even ensemble-adversarially trained models are susceptible to unrestricted filter-based adversarial perturbations. This indicates the potential of the proposed method whose goal is to transform image filters into malicious applications.

Table 1: Attack Success Rate (ASR)

| | Multi-network | | | Single-network | | |
|---|---|---|---|---|---|---|
| **Setup** | **5-NET** | **4-NET** | **3-NET** | **MobileNet** | **ResNet50** | **VGG19** |
| **3filters** | 44% | 50 % | 52% | 64% | 74% | 70% |
| **4filters** | 58% | 50% | 70% | 74% | 88% | 82% |
| **5filters** | 68% | 66% | 76% | 78% | 96% | 86% |

**Comparisons with similar attack models.**

We also compare our algorithm with other existing adversarial attack methods that have similar colorization and filtering characteristics.

**SemanticAdv** (Hosseini & Poovendran, 2018) randomly changes the hue and saturation values of an image while not affecting the shape of objects.
**Colorfool** (Shahin Shamsabadi et al., 2020) uses image semantics to modify the colors by employing priors on color perception trying. It tackles the limitations of SemanticAdv which has been found to produce unnatural looking images.
**Edgefool** (Shamsabadi et al.) and **FilterFool** (Shahin Shamsabadi et al., 2021) generate adversarial examples by applying generic image-enhancement filters obtained by means of neural networks. For example, FilterFool-GC tries to create a perturbation that imitates the gamma correction filter. It is important to note that these methods do not work in a pure hard-label black-box setting because they use the logits information that in general are not exposed and visible to the final user.
**ACE** (Zhao et al., 2020), **ReColorAdv** (Laidlaw & Feizi, 2019), and **cAdv** (Bhattad et al., 2019) methods use gradient information in order to optimize a color transformation.
In particular, we focus on **ACE-Ins1** and **ACE-Ins2** which employ two Instagram filters to guide the color optimization towards specific styles of enhancement, namely Nashville and Toaster. Adversarial samples generated by the above mentioned methods are reported in Appendix A.

Table 2: Transferability rates

| | Trained on | | | | | | | | |
|---|---|---|---|---|---|---|---|---|---|
| | **5-NET** | | | **4-NET** | | | **3-NET** | | |
| **Tested on** | 3f | 4f | 5f | 3f | 4f | 5f | 3f | 4f | 5f |
| **InceptionV3** | 0.636 | **0.655** | 0.617 | 0.600 | 0.517 | 0.575 | 0.500 | 0.457 | 0.578 |
| **InceptionV4** | 0.545 | **0.793** | 0.705 | 0.600 | 0.551 | 0.757 | 0.538 | 0.657 | 0.657 |
| **IncResV2-ens-adv** | 0.545 | **0.655** | 0.558 | 0.480 | 0.517 | 0.636 | 0.269 | 0.514 | 0.447 |
| | Trained on | | | | | | | | |
| | **MobileNet** | | | **ResNet50** | | | **VGG19** | | |
| **Tested on** | 3f | 4f | 5f | 3f | 4f | 5f | 3f | 4f | 5f |
| **InceptionV3** | 0.375 | 0.378 | **0.410** | 0.216 | **0.340** | 0.312 | 0.257 | **0.365** | 0.348 |
| **InceptionV4** | 0.437 | 0.540 | **0.564** | 0.405 | 0.477 | **0.541** | 0.342 | 0.439 | **0.441** |
| **IncResV2-ens-adv** | 0.250 | 0.297 | **0.359** | 0.216 | 0.250 | **0.270** | **0.285** | 0.243 | 0.279 |

Table 3: Comparison with similar colorization and filtering techniques

|  | AGV-5-NET 5f | AGV-single 5f | SemanticAdv | Colorfool | Edgefool |
|---|---|---|---|---|---|
| **ASR-ref** | 68% | 96% | 88 % | 91.7% | 98.1% |
| **avg-TR** | 62.75 % | 37.5% | 65.45% | 47.00% | 43.45% |
| **min-TR** | 55.88 % | 27.08% | 54% | 34.8% | 35.7% |
|  | (IRV2-ens-adv) | (IRV2-ens-adv) | (Resnet18) | (Resnet18) | (Resnet18) |
|  | **Filterfool GC** | **ACE-Ins1** | **ACE-Ins2** | **ReColorAdv** | **cAdv** |
| **ASR-ref** | 99.9% | 99.3% | 99.4 % | 89.2% | 99.83% |
| **avg-TR** | 34.7% | 24.24% | 26.31% | 17.79% | 29.78% |
| **min-TR** | 29.2% | 11.47% | 9.9 % | 10.77% | 28.56% |
|  | (Resnet18) | (IncV3) | (IncV3) | (Dense121) | (Dense121) |

In Table 3 we report the Attack Success Rates obtained by crafting adversarial samples on the reference model (ASR-ref) and the average and minimum Transferability Rate (avg-TR, min-TR) with respect to different testing models. The reference model is ResNet50 for all the methods, with the exception of ReColorAdv that used InceptionV3. For the minimum values we also report the target network which represents the hardest model to attack. The experiments show that in most cases our algorithm achieves better results than the other techniques while using less information (top1-label vs logits).

From the ASR point of view, the AGV-single method performs better than SemanticAdv and Colorfool but it falls behind on the transfer rate, while AGV-multi-network outperform almost all other systems on the hold-out models. Given the random nature of its approach, the SemanticAdv generates suspicious perturbations that lead to excessively transformed images (people with blue skin or green horses) which can induce misclassification in unseen models more easily. Even though Colorfool tries to limit the adversarial perturbations by using image segmentation to modify the colors within specific ranges, it often produces noticeable, artificial colorization effects. This is mostly due to the inaccuracy of the image segmentation techniques that poorly identifies the objects resulting in unnatural colorization patterns, such as skies with pink clouds or dogs half white, half blue. Despite accessing only the final class-label decision of a model, AGV reaches a higher transferability rate than many other methods relying on richer information for the attack process, such as FilterFool, ACE, ReColorAdv and cAdv. Moreover, we have a great diversity in terms of filter styles, ranging from more bright and colourful to more soft and subtle transformations which are frequently found on many social media platforms, like Instagram. Examples that sucessfully transfer to the InceptionResNetV2-ens-adv are illustrated in Figure 2 and 3 in Appendix A.

## 4.4 STEALTHINESS OF ADVERSARIAL EXAMPLES

We evaluate the stealthiness of the adversarial examples from two perspectives: stealthiness to human perception and stealthiness to defense mechanisms. We measured the human perception by means of most common No-Reference image quality assessment metrics (NR-IQA), such as NIMA, NIQE and BRISQUE (Talebi & Milanfar, 2018; Mittal et al., 2012; 2013). For this analysis, we calculated the average of the normalized differences between the NR scores on the original images and the corresponding modified images. The results show no significant difference between the scores of clean images and adversarial samples obtained by applying the filter sequence found by the algorithm. Considering the average computed over 12 different experiments that produced about 60k adversarial images, we obtained values in $[-0.0479, 0.0006]$, with an average of $-0.0144$.

For the stealthiness to defense mechanisms we considered well-known detecting and sanitizing techniques proposed to counter adversarial attacks. As detection methods we tested Feature Squeezing (FS) (Xu et al., 2018) and detectors based on autoencoders and model distillation available in the Alibi Detector library (Van Looveren et al., 2019). Experimental results on CIFAR-10 dataset show that FS is able to detect less than 10% of adversarial examples as malignant. The autoencoder and distillation based methods were able to reach only 25% and 11% detection rate, respectively. The adversarial threshold was inferred by assuming 5% of the clean images as outliers. As sanitizing

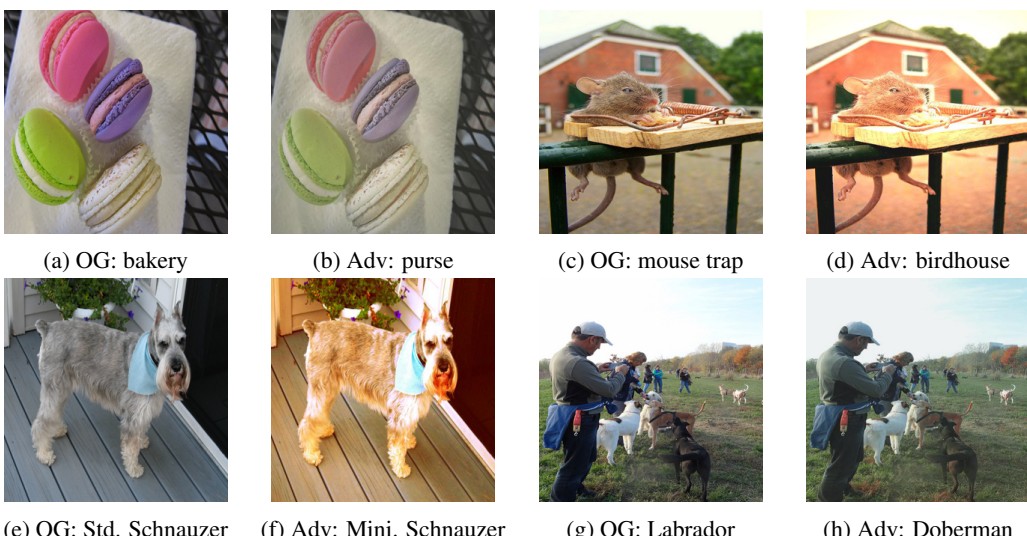

(a) OG: bakery     (b) Adv: purse     (c) OG: mouse trap     (d) Adv: birdhouse

(e) OG: Std. Schnauzer     (f) Adv: Mini. Schnauzer     (g) OG: Labrador     (h) Adv: Doberman

Figure 2: Examples of clean images and their corresponding adversarial example. The adversarial labels are obtained by transferring the modified images to the InceptionResNetV2-ens-adv model.

techniques we tested jpeg compression, Destylization (Wu et al.) and Instragram-Filter-Removal-Network (Kinli et al., 2021). Preliminary tests indicate that these methods cannot effectively remove the adversarial perturbations produced by combining multiple filters. In the case of jpeg compression we obtain, in most cases, an increasing number of attacks.

### 4.5 DECEPTION ABILITY

An effective attack can be evaluated also with respect to its deception ability, that can be measured by the distance between the original and the assigned class after the modification. To do this we computed the rank of the class of the transformed image inside the class distribution of the original image (descending order with respect to the prediction probabilities): the further away, the stronger the deception is. Similarly, we also examined the rank of the right class in the distribution of the modified images in order to check how far away it was moved.

Preliminary results showed that more than 38% of the modified images have been classified with a class that in the original distribution was, on average, over the 5th place. On the other hand, in more than 28% the right class was placed, on average, over the 5th position in the distribution computed for the filtered images.

## 5 CONCLUSIONS AND FUTURE WORKS

In this work we present the algorithm AGV-multi-network for the optimization of Instagram-style image filters in a multi-network setting able to fool classification systems with a low cost in term of queries and a high transferability rate.

The algorithm was experimentally tested on a subset of the ImageNet validation set. From the testing phase we can conclude that: (*i*) the algorithm shows very good transferability rate also in the case of adversarial trained network and with respect to other method producing images with evident artifacts; (*ii*) the algorithm is able to craft the attacks using a very limited number of queries.

There is a big room for improvements that are under study. In particular we aim to exploit the multi-objective fitness function of the original AGV algorithm in order to craft targeted deceptions, move the right classification even further away in the modified class distribution, apply automatic measure for image quality assessment and optimize the attacks also to avoid detection methods.

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

## A   APPENDIX

We report different adversarial examples generated by state-of-the-art methods and similar systems.

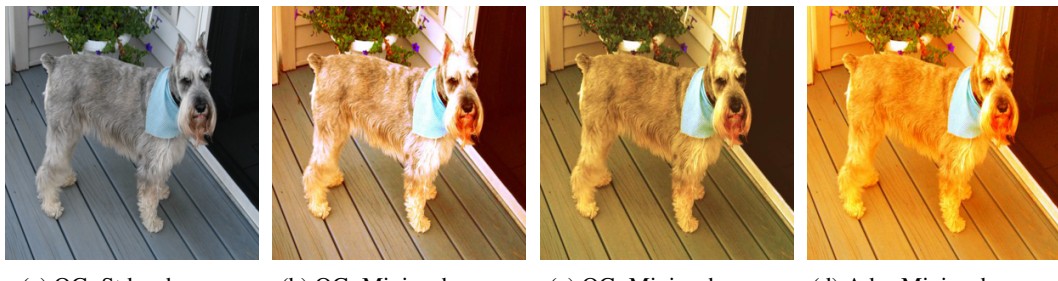

(a) OG: Std. schnauzer     (b) OG: Mini. schnauzer     (c) OG: Mini. schnauzer     (d) Adv: Mini. schnauzer

Figure 3: Examples of different adversarial perturbations styles crafted on the same image by AGV-multi-network with different configurations.

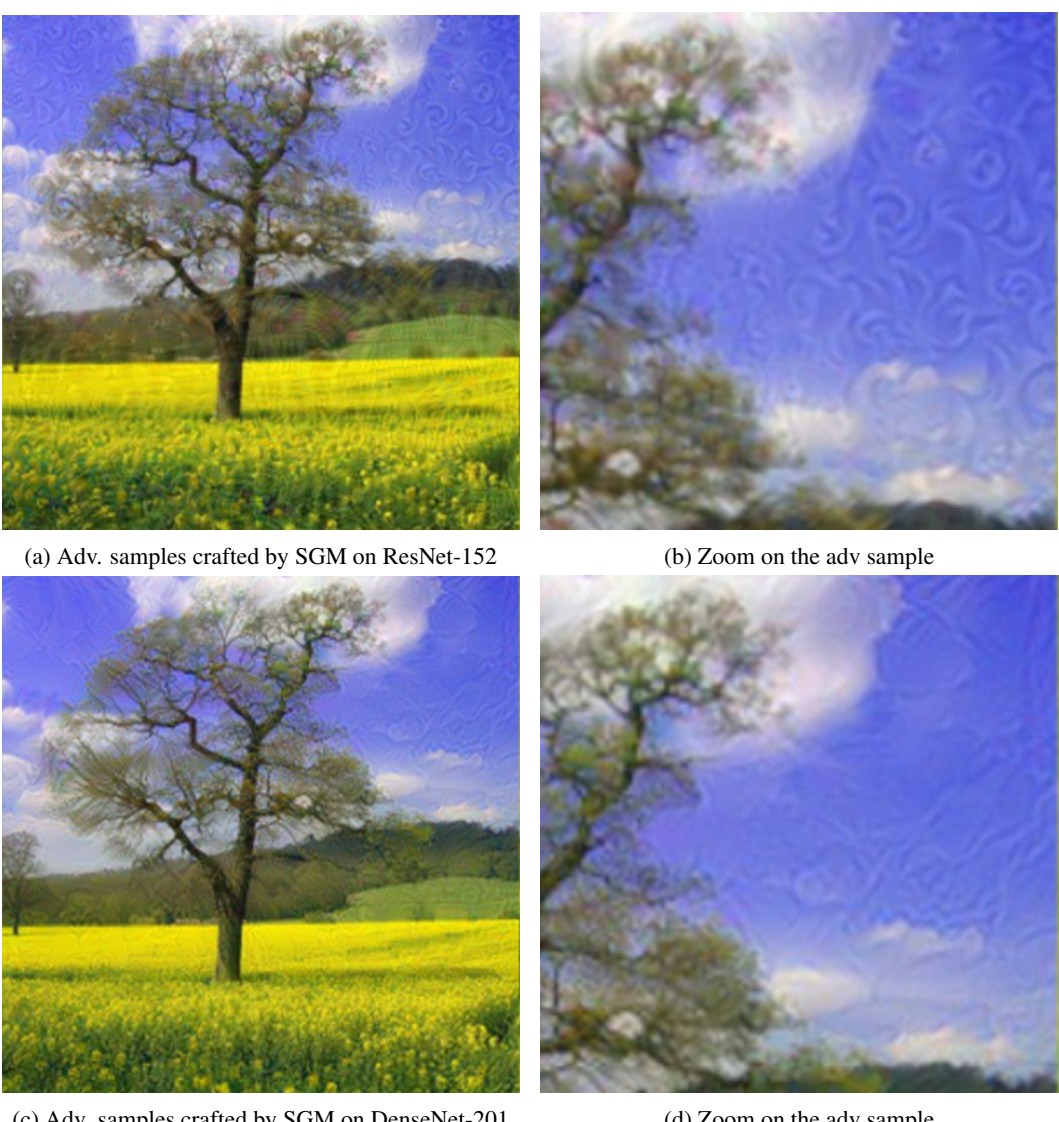

(a) Adv. samples crafted by SGM on ResNet-152      (b) Zoom on the adv sample

(c) Adv. samples crafted by SGM on DenseNet-201      (d) Zoom on the adv sample

Figure 4: Advesarial samples crafted by SGM (Wu et al., 2020) method. The adversarial perturbations creates visible artifacts in the image.

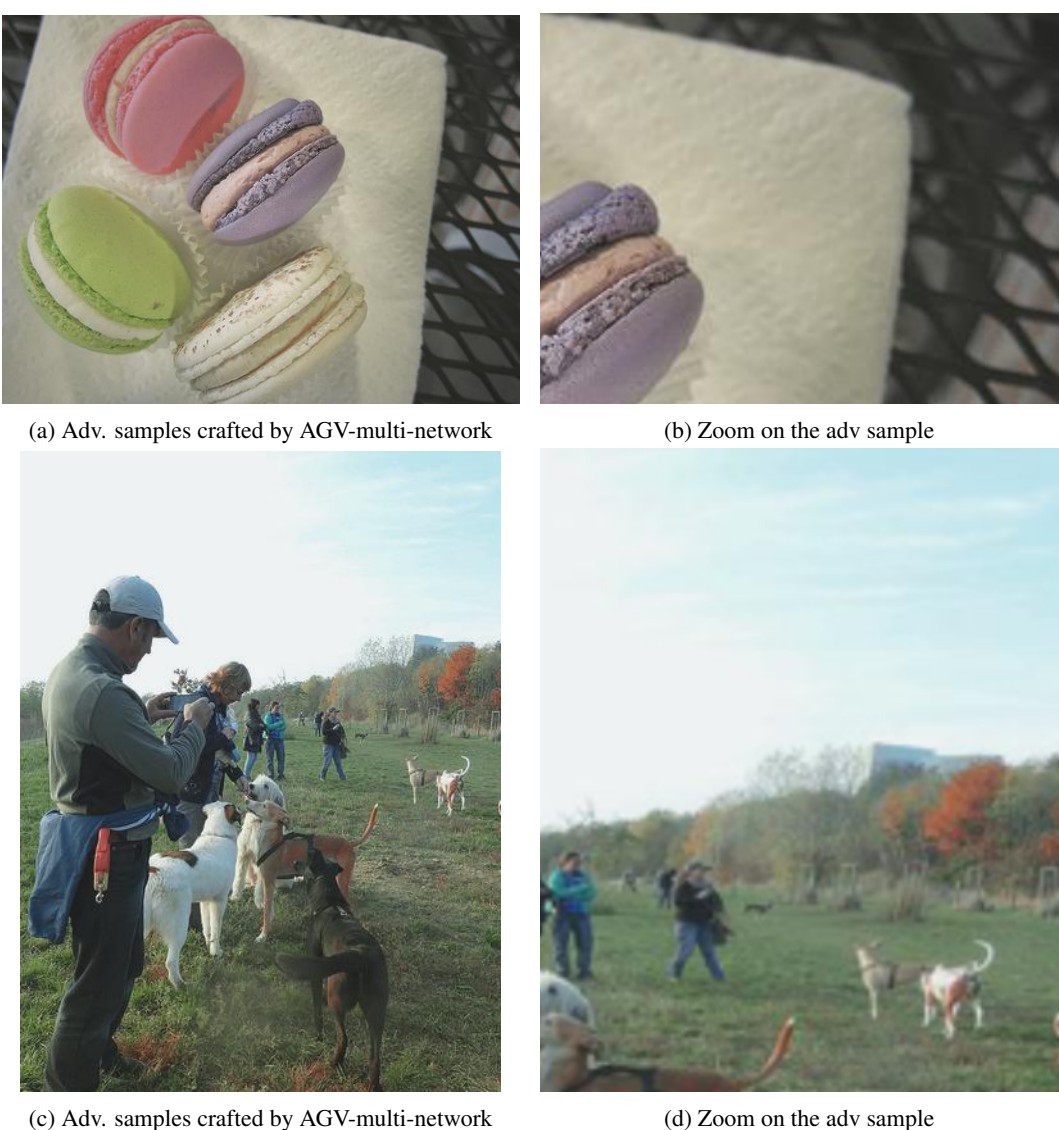

(a) Adv. samples crafted by AGV-multi-network      (b) Zoom on the adv sample

(c) Adv. samples crafted by AGV-multi-network      (d) Zoom on the adv sample

Figure 5: Examples of adversarial samples generated by AGV-multi-network. The adversarial perturbation does not create artificial patterns in the image.

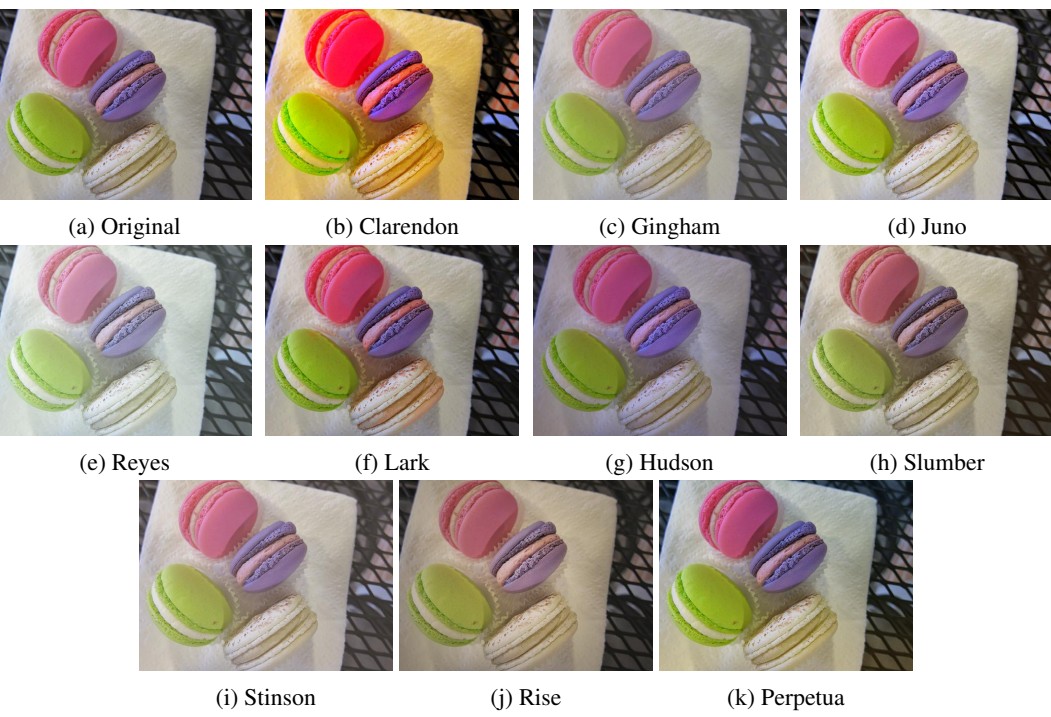

(a) Original     (b) Clarendon     (c) Gingham     (d) Juno

(e) Reyes     (f) Lark     (g) Hudson     (h) Slumber

(i) Stinson     (j) Rise     (k) Perpetua

Figure 6: The effects produced by applying filters individually with default paramters.

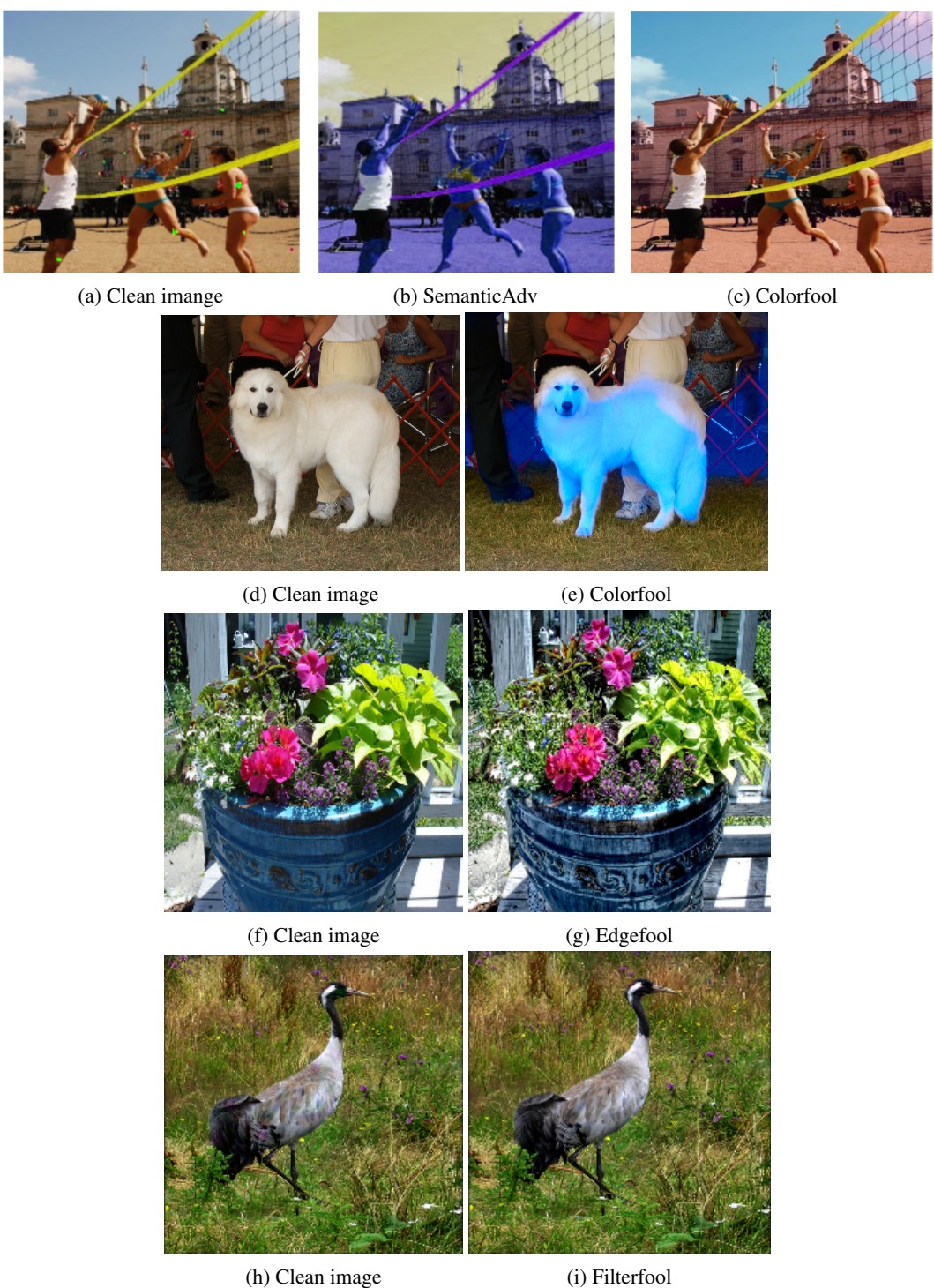

Figure 7: Adversarial examples generated with other filter-based and colorization systems.

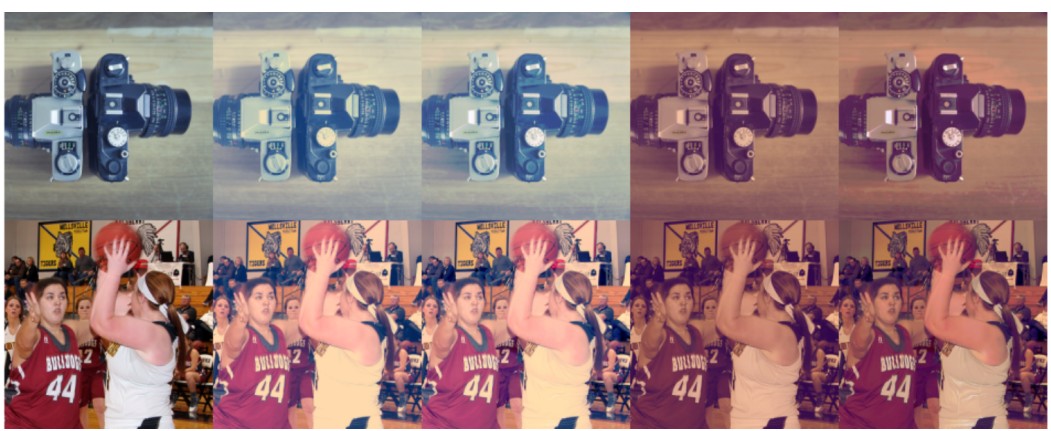

(a) ACE-Ins with color style guidance from Instagram filters

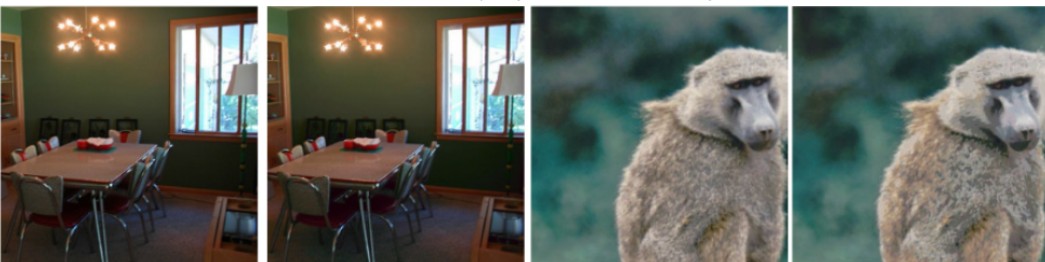

(b) ReColorAdv adversarial examples

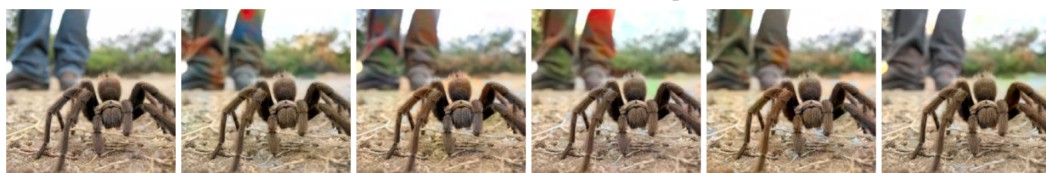

(c) cAdv adversarial examples

Figure 8: Adversarial examples generated with other filter-based and colorization systems.

