# OpenReview forum: "One for Many: an Instagram inspired black-box adversarial attack"
_ICLR.cc/2022/Conference — ICLR 2022 Submitted_

### Official Review · Reviewer_1MTN · 2021-11-01

**Correctness:** 3
**Technical Novelty And Significance:** 3
**Empirical Novelty And Significance:** 2
**Recommendation:** 5
**Confidence:** 4

**Main Review:**

Strengths:
This paper was clearly written and well organized with interesting motivations. Such Instagram-style filters are commonly used in daily life and might easily achieve lower suspicion than conventional techniques (large l_p perturbations). In addition, the authors compared a lot of other "unrestricted attacking" methods and demonstrated AGV can achieve higher transferability with less queries.

Weakness:
Experiments are not complete.

(1) The paper claimed, "our goal is to find one adversarial perturbation per-image that can fool many deep learning models". Therefore, when evaluating the performance of methods, it is necessary to present the worst performance on one model besides the average one. (Table 3) Readers can learn what model structures are harder for unconstrained adversarial attacks.

(2) It is unclear to me that the comparison between different methods is fair. In table 3, it seems the objective of AGV-5-NET 5f is to attack 5 different models. But I am not sure if other methods are also trained for attacking 5 different models.

(3) In table 2, the test model is limed to InceptionV3, InceptionV4, adversarially trained InceptionResnetV2-ens-adv [1], one potential improvement for this paper is to test more model architectures (e.g. ViTs). Also, details of InceptionResNetV2-ens-adv are missing (how is it adversarially trained?).

(4) I am not sure how to replicate this paper. To be specific, how does the inner algorithm in 3.3.2 perturb the original image? What is the initial learning rate and decay rate in the inner algorithm? I highly recommend the authors upload the codes or include more details for the methods.



Visual examples are limited.
The authors claimed, "our filter composition cannot be distinguished from any other filter composition used extensively every day to enhance images". However, the paper only includes several visual examples to demonstrate that. I am not sure the proposed methods can achieve low suspiciousness without any human studies or quantitative analysis. In addition, ten image filters used in the paper should also be visualized.

I think the paper can be largely improved if previous issues are solved.

Questions:
How does the proposed method perform in terms of computational efficiency compared to others?
Is it possible to control the tradeoff between effectiveness and distortion of the image (e.g. constraining the parameters)?

Other comments/suggestions:
The authors should add a section of related works and summarize some important papers in the adversarial ML community e.g. [2,3,4,5].

[1] Ensemble adversarial training: Attacks and defenses, Tramer et al., ICLR 2018

[2] Explaining and harnessing adversarial examples, Goodfellow et al., ICLR 2015

[3] Intriguing properties of neural networks, Szegedy et al., ICLR 2014

[4] Adversarial examples in the physical world, Kurakin et al., CoRR 2016

[5] Transferability in machine learning: from phenomena to black-box attacks using adversarial samples, Papernot et al., arXiv 2016


**Summary Of The Paper:**

This paper proposed a nested evolutionary algorithm to generate adversarial perturbations under the black-box settings. Such perturbations are composed of various image filters inspired by Instagram and can simultaneously attack multiple neural networks. They claimed that the attacks were semantically robust and had a low cost of queries.


**Summary Of The Review:**

The idea in this paper is very interesting and novel to the adversarial machine learning community, however, issues on experiments and presentations prevent me from accepting it.

---

> ### Author Response · Authors · 2021-11-16
> **Response to Reviewer 1MTN (part 2)**
>
> **Visual examples are limited. The authors claimed, "our filter composition cannot be distinguished from any other filter composition used extensively every day to enhance images". However, the paper only includes several visual examples to demonstrate that. I am not sure the proposed methods can achieve low suspiciousness without any human studies or quantitative analysis. In addition, ten image filters used in the paper should also be visualized.**
>
> The low suspiciouness of the attacks is one of  our main targets. An extensive experimental phase on the filters have been conducted in order to choose appropriate domains for the values of  $\alpha$ and $s$. These constraints allow the filters to produce controlled modifications that, otherwise, would modify too much the image.
> The effects produced by the ten filters used by the algorithm are reported in Fig.6 of the Appendix. Filters with default parameters $\alpha=1$ and $s=1$ are shown.
>
> Moreover we added the new Section 4.4 STEALTHINESS OF ADVERSARIAL EXAMPLES where some results of our studies about image quality assessment are reported.
>  We tested the most common No-Reference measures like
>     NIMA, NIQE, and BRISQUE and we evaluated how the values of these metrics change after the filter applications.  For this analysis, we calculated the average of the normalized differences between the NR scores on the original images and the corresponding modified images. The results show no significant difference between the scores of clean images and adversarial samples obtained by applying the filter sequence found by the algorithm. Due to the lack of space we reported just some summarized results. Considering the average computed over 12 different experiments that produced about 60k adversarial images, we obtained values for the mean of the normalized differences of the IQA metrics ranging in $[-0.0479, 0.0006]$, with an average of $-0.0144$.
>
> Finally we can say that we are working on a Multi-Objective version of the system that should include image quality assessment in the objective function of the  evolutionary algorithm.
>
> **How does the proposed method perform in terms of computational efficiency compared to others?**
>
> The computational efficiency of the proposed method directly depends on the number of queries to the target model that are necessary to craft the attack.
> As you can note in Section 3.4.1 the average number of queries is (relatively) low, and it is lower than other Black-box approaches.
> In order to explain better this point we added, in the new version of the paper, the section 4.2 "Query Efficiency" where some comparisons with state of the art methods are reported.
>
> Just to give an idea about the time used: the version MultinetworkAGV-5Net-5filters  takes about 50s to make an external generation querying all the 5 victim models (160 queries for each model) with the experiment configuration reported in the paper. The experiments run on a GPU RTX3090.
>
> Moreover, it should be noted that the idea of crafting adversarial examples with higher transferability rate works exactly in the direction of efficiency. Even if the crafting process would require more resources, then, when already crafted attacks can be reused, the attacking phase is quicker and more efficient.
>
> **Is it possible to control the tradeoff between effectiveness and distortion of the image (e.g. constraining the parameters)?**
>
> We think that a reply to this comment is included and provided in the previous two replies.
>
> **Other comments/suggestions: The authors should add a section of related works and summarize some important papers in the adversarial ML community e.g. [2,3,4,5].**
>
> We are aware that the suggested references are pillars of adversarial machine learning and there surely must be included in a paper with a wide bibliography. For this work, due to the limited space, we preferred to cite more recent papers or papers more directly related to our work.
>
>
> [2] Explaining and harnessing adversarial examples, Goodfellow et al., ICLR 2015
>
> [3] Intriguing properties of neural networks, Szegedy et al., ICLR 2014
>
> [4] Adversarial examples in the physical world, Kurakin et al., CoRR 2016
>
> [5] Transferability in machine learning: from phenomena to black-box attacks using adversarial samples, Papernot et al., arXiv 2016

---

> ### Author Response · Authors · 2021-11-16
> **Response to Reviewer 1MTN (part 1)**
>
> We would like to thank the reviewer for the valuable feedback. Below are our response to the comments.
>
> **The paper claimed, "our goal is to find one adversarial perturbation per-image that can fool many deep learning models". Therefore, when evaluating the performance of methods, it is necessary to present the worst performance on one model besides the average one. (Table 3) Readers can learn what model structures are harder for unconstrained adversarial attacks.**
>
> Results reported in the first row "ASR-ref" of Table 3 represent the success rate of the attacks when build against the reference model. In our case the reference model is the set of the 5 networks described in 4.1 "Experimental Setup" and the results in the Table indicate the percentage of the images in the dataset that could be successfully modified to create an attack that is able to  simultaneously fool all the 5 networks. It is not an average with respect the 5 networks. In 68\% of cases it is possible to build such an attack. This is the motivation why this value is sensibly lower than the corresponding values for all the compared methods that built the attack with just a single network as objective.
>   On the other hand, results reported in row "avg-TR" of Table 3 show the average values with respect to the attacked models used to test the transferability of the attacks, so models that are different from the ones used as reference during the crafting phase.
>   In this case it is possible to show also the minimum and Table 3 has been accordingly updated with the new row "min-TR".
>   We would just point out that this information does not necessarily  help the reader to establish which model is harder to be attacked. This is  because the set of  models used by the respective authors to test the transferability is always different among the models. So, the model that results harder for an algorithm probably is never tested by the others (the results have been taken from the original papers and are not recomputed because of computational and reimplementation efforts). This is also the motivation that induced us to report the average values with respect the different models.
>
> **It is unclear to me that the comparison between different methods is fair. In table 3, it seems the objective of AGV-5-NET 5f is to attack 5 different models. But I am not sure if other methods are also trained for attacking 5 different models.**
>
> We think that the possibility to easily include more networks is one of the strengths of the method we propose with respect to other methods that are not designed to do so. The most fair comparison should be against black box models able to manage more networks (via an ensemble or any other technique). The problem is that, to the best of our knowledge, similar systems do not exist. Available systems we know work only in a white box setting, such as (Liu et at.2017, Xie et at.2019, Dong et al. 2019).
>
> So, we decided that the best fair comparison was against other black box, colorization or filter based methods that share with us either the main setting or the typology of attack.
>
> **In table 2, the test model is limed to InceptionV3, InceptionV4, adversarially trained InceptionResnetV2-ens-adv [1], one potential improvement for this paper is to test more model architectures (e.g. ViTs).**
>
> Yes, we totally agree with the referee. We are working on the extension of the tested models. We greatly appreciated the suggestion about ViTs that surely represent an interesting challenge.
>
> **Also, details of InceptionResNetV2-ens-adv [1] are missing (how is it adversarially trained?)**
>
> We are sorry, but due to the lack of space it was not possible to add the details about the adversarial training. The reader should refer to the original paper.
>
> [1] Ensemble adversarial training: Attacks and defenses, Tramer et al., ICLR 2018
>
> **I am not sure how to replicate this paper. To be specific, how does the inner algorithm in 3.3.2 perturb the original image? What is the initial learning rate and decay rate in the inner algorithm? I highly recommend the authors upload the codes or include more details for the methods.**
>
> All the parameters for the inner algorithm are reported in the last paragraph of Section 4.1 "EXPERIMENTAL SETUP - Implementations details". Moreover, we plan to make the code publicly available after the double-blind review process.

---

### Official Review · Reviewer_Cohx · 2021-11-02

**Correctness:** 4
**Technical Novelty And Significance:** 3
**Empirical Novelty And Significance:** 3
**Recommendation:** 6
**Confidence:** 4

**Details Of Ethics Concerns:**

New adversarial attack is presented with all corresponding concerns.


**Main Review:**

Strengths:
1) Interesting evolutionary strategy to craft strong transferable attacks using image filters.
2) Adversarial examples look only as filtered clean images with no artificial patterns as in other gradient based methods.

Weaknesses:
1) Potentially weak attack. Judging by the examples shown in the paper, the adversarial images are mislabeled to a class which is very close to the ground truth (for instance, from “standard schnauzer” to “mini schnauzer” or from “Labrador” to “Dobermann” ) indicating that the attack strength is low. Is there any way to fix this? Can you make an attack more semantically strong?
2) Minor grammatical errors (e.g. “simoultaneously”)


**Summary Of The Paper:**

The paper proposes a new way to craft adversarial examples using Instagram inspired filters by means of evolutionary algorithms. The resulting adversarial attacks do not have recognizable artificial patterns as in other methods and show competitive performance including a difficult black-box scenario.


**Summary Of The Review:**

Although the attack looks indistinguishable from just a filtered image, it seems that the strength of the attack is questionable, in most cases images are mislabeled to a class which is very close to ground truth. Unfortunately, it undermines the whole purpose of the method, its practicality and effectiveness. Therefore, I tend to be on the rejection side.

---

> ### Author Response · Authors · 2021-11-16
> **Response to Reviewer Cohx**
>
> We thank the reviewer for taking time to review our work.
>
> **Potentially weak attack. Judging by the examples shown in the paper, the adversarial images are mislabeled to a class which is very close to the ground truth (for instance, from “standard schnauzer” to “mini schnauzer” or from “Labrador” to “Dobermann” ) indicating that the attack strength is low. Is there any way to fix this? Can you make an attack more semantically strong?**
>
> Thanks to this comment we realized that the images used could be misleading and could induce wrong deductions. In order to avoid this, we extend the paper with a new section 4.5 "Deception ability" that contains a summarization of preliminary tests we made about the average ability of the system to move the new (wrong) classification away from the correct one.
> Moreover, in order to understand why the reported images can give the impression of a low significance of the attacks, the reviewer has to consider the ImageNet unbalancing towards given classes, especially the ones related to the dog breeds: ImageNet contains 90 categories for dog breeds and a huge amount of images about dogs.
> With a random selection, it is very common to extract images about dogs.
> If the reviewers think that images about dogs could be misleading we can change them with others.

---

> > ### Comment · Reviewer_Cohx · 2021-11-29
> > **Thank you for the answer!**
> >
> > I believe the authors addressed my concern about the strength of the proposed attack. Thus, I increase my score.

---

### Official Review · Reviewer_eN8o · 2021-11-02

**Correctness:** 3
**Technical Novelty And Significance:** 2
**Empirical Novelty And Significance:** 2
**Recommendation:** 3
**Confidence:** 4

**Main Review:**

# Strengths
- Addressing an important problem: hard-label black-box setting is a more practical attack setting compared to existing white-box or soft-label black-box attacks.
- Combining evolutionary algorithm and model ensembling to achieve good black-box attack and transferability performance.


# Weaknesses
- The intellectual delta from existing works is low. As also identified by the authors, the idea of using image filters is not new--many existing works (SemanticAdv, Colorfool, Edgefool, Filterfool, ACE, etc) have already adopted such a concept to generate benign-looking AEs. The difference is that prior works assume knowledge on the confidence scores or even model internals. The authors address this by using evolutionary algorithms to find the best filter parameters, which is also a straightforward solution and the results are not surprising. To improve this, the authors may consider comparing different evolutionary algorithms such as GA, PSO, to systematically understand the tradeoff between the different methods.
- No evaluation on the AE stealthiness. Since the benefit of using image filters are the attack stealthiness as the generated AEs are intended to look benign. However, I cannot find any stealthiness evaluation in the paper. There are only a few images of the generated AEs shown in the paper. But that is not a systematic evaluation. There are two directions to address this. One solution is to calculate image perceptual metrics (such as those used in GANs [1]) of the AE. Another is to conduct a user study on the AEs to ask participants about the suspiciousness. For both solutions, random sampling of the successful AEs is required to ensure no biases in the selection.
- The good transferability mostly comes from the model ensembling as can be seen from the much lower transferability of AVG-single 5f in Table 3.

[1] An empirical study on evaluation metrics of generative adversarial networks, https://arxiv.org/pdf/1806.07755.pdf

**Summary Of The Paper:**

The paper proposes a black-box attack method that uses an evolutionary algorithm to find the best image filter parameters that can achieve untargeted attacks. Model ensembling is also used in the AE generation to improve attack transferability. The proposed method is compared to other similar image filtering-based AE generation methods. Results show that the proposed method is able to outperform most of the related works in terms of transferability.

**Summary Of The Review:**

The paper studies an important problem. However, the current evaluation in the paper is still lacking. More systematic evaluations on the evolutionary algorithms and stealthiness are necessary.

---

> ### Author Response · Authors · 2021-11-16
> **Response to Reviewer eN8o**
>
> We would like to sincerely thank you for your detailed feedback. We will try to address your concerns about our work below.
>
> **The difference is that prior works assume knowledge on the confidence scores or even model internals. The authors address this by using evolutionary algorithms to find the best filter parameters, which is also a straightforward solution and the results are not surprising. To improve this, the authors may consider comparing different evolutionary algorithms such as GA, PSO, to systematically understand the tradeoff between the different methods.**
>
> We agree with the reviewer's suggestion of extending the set of evolutionary algorithm used. During this study we considered other approaches for the internal algorithm such as GA and hyperparameter optimization frameworks (in particular Optuna and SMAC3). Preliminary experiments showed that ES offered the best trade-off between attack effectiveness and computational effort. For example Optuna achieved slightly higher attack success rate but it required more time. We did not report results for lack of space. If the reviewers think that they could be interesting we can provide a table with the results in the paper appendix.
> The extension to other techniques for the external algorithm will be surely taken into consideration for future works.
>
> **No evaluation on the AE stealthiness. Since the benefit of using image filters are the attack stealthiness as the generated AEs are intended to look benign. However, I cannot find any stealthiness evaluation in the paper. There are only a few images of the generated AEs shown in the paper. But that is not a systematic evaluation. There are two directions to address this. One solution is to calculate image perceptual metrics (such as those used in GANs [1]) of the AE. Another is to conduct a user study on the AEs to ask participants about the suspiciousness. For both solutions, random sampling of the successful AEs is required to ensure no biases in the selection.**
>
> This observation  hits one of the key points of our work that we did not include in the first version of the paper due to lack of space.
>     Now we added the new section 4.4 "STEALTHINESS OF ADVERSARIAL EXAMPLES".
>     In practice we have to manage two kinds of stealthiness: stealthiness to human perception
>     and stealthiness to attack detectors.
>     We made some experiments to evaluate the stealthiness to human perception by means of image quality assessment metrics. We tested the most common No-Reference measures like
>     NIMA, NIQE, and BRISQUE and we evaluated how the values of these metrics change after the filter applications.  For this analysis, we calculated the average of the normalized differences between the NR scores on the original images and the corresponding modified images. The results show no significant difference between the scores of clean images and adversarial samples obtained by applying the filter sequence found by the algorithm. Considering the average computed over 12 different experiments that produced about 60k adversarial images, we obtained values in $[-0.0479, 0.0006]$, with an average of $-0.0144$.
>
>    From the point of view of stealthiness to defense techniques, we considered both detecting and sanitizing mechanisms that have been proposed in literature to work against different attacks and specialized for attacks crafted by means of Instagram-inspired image filters. As detection methods we tested Feature Squeezing (FS) [4] and detectors based on autoencoders and model distillation available in the Alibi Detector library [5]. Experimental results on CIFAR-10 dataset show that FS is able to detect less than 10\% of adversarial examples as malignant. The autoencoder and distillation based methods were able to reach only 25\% and 11\% detection rate, respectively. The adversarial threshold was inferred by assuming 5\% of the clean images as outliers. As sanitizing techniques we tested jpeg compression, Destylization [2] and Instragram-Filter-Removal-Network [3]. Preliminary tests indicate that these methods cannot effectively remove the adversarial perturbations produced by combining multiple filters.
>
>
> [1] An empirical study on evaluation metrics of generative adversarial networks, https://arxiv.org/pdf/1806.07755.pdf
>
> [2] Recognizing Instagram Filtered Images with Feature De-stylization. 2020,  https://arxiv.org/pdf/1912.13000.pdf
>
> [3] Instagram Filter Removal on Fashionable Images, https://arxiv.org/pdf/2104.05072.pdf
>
> [4]  Feature Squeezing: Detecting Adversarial Examples in Deep Neural Networks, https://arxiv.org/pdf/1704.01155.pdf
>
> [5] Alibi detect, https://docs.seldon.io/projects/alibi-detect/en/stable/overview/getting_started.html

---

### Official Review · Reviewer_PWpF · 2021-11-04

**Correctness:** 3
**Technical Novelty And Significance:** 2
**Empirical Novelty And Significance:** 3
**Recommendation:** 6
**Confidence:** 2

**Main Review:**

**Strengths**
1. An interesting research direction is explored and natural-looking attacks are proposed.
2. The proposed family of attacks is a black box and more applicable in the real world.
3. The attacks seem to use fewer queries making them more effective.
4. Comprehensive set of experiments are performed to show the effectiveness.


**Weaknesses**
1. One interesting thing is that the attacks produce natural-looking artifacts. However, it would be interesting to see some fail cases.
2. A comparison of the number of queries used by these attacks vs. other black-box attacks should be added.

**Additional Recommendation**
1. It would be interesting to see if Instagram filters can also be used to improve privacy, something similar to [1].

References

[1] Unlearnable Examples: Making Personal Data Unexploitable


**Summary Of The Paper:**

This paper introduces a new family of black-box adversarial attacks. These attacks are constructed by composing Instagram filters-based transformation in the input space. Perturbations in the input space are unrestricted and large but only produce natural-looking artifacts. The input is transformed with 10 different filters. Each filter has two variables: alpha to control the intensity and s to control strength. To get the optimal values of these parameters, an evolutionary approach is used. Comprehensive experiments are performed to show the efficacy of the attacks.

**Summary Of The Review:**

This paper introduced an interesting set of black-box attacks and showed their utility through comprehensive experiments. It seems like a good paper. However, I am not very familiar with recent work on black-box attacks.

---

> ### Author Response · Authors · 2021-11-16
> **Response to Reviewer PWpF**
>
> We would like to thank the reviewer for the insightful review. Below are our response to the comments.
>
> **A comparison of the number of queries used by these attacks vs. other black-box attacks should be added.**
>
> Direct comparisons with other methods are in general difficult due to the differences of the approaches, especially in the case of $L_p$-bounded modifications. Anyway, we added a new section 4.2 "Query Efficiency" in the new version of the paper where some comparisons with other recent, query-efficient and black-box attacks are summarized.
>
> **It would be interesting to see if Instagram filters can also be used to improve privacy, something similar to [1].**
>
> Privacy preserving application is one of the key factors that motivated us to transform the universal approach proposed in (Baia et al 2021) into a per-instance approach. In this way we can produce image specific modifications that are more suitable in the context of social media, like Instagram.
> In this case the user privacy protection is realized by attacking social media information extraction applications based on Deep Learning models.
>
> Regarding the reference [1], it indeed seems very interesting for our purposes, but unfortunately we were not able to find any paper with this (or any other similar) title. Could you provide additional information or a link to it?
>
>  $[1]$ Personal privacy: Make your social media images unusable for AI}

---

> > ### Comment · Reviewer_PWpF · 2021-11-16
> > **Modified paper title**
> >
> > Sorry for the wrong reference. I have updated the reference. The paper I mentioned is: "Unlearnable Examples: Making Personal Data Unexploitable" available at this [link](https://openreview.net/forum?id=iAmZUo0DxC0).

---

> > > ### Author Response · Authors · 2021-11-17
> > > **Response to Reviewer PWpF for the paper recommendation**
> > >
> > > Thank you for this recommendation and for providing the link. The paper is very inspiring and interesting. We will definitely take it into consideration for future works.

---

### Official Review · Reviewer_chwU · 2021-11-04

**Correctness:** 3
**Technical Novelty And Significance:** 2
**Empirical Novelty And Significance:** 2
**Recommendation:** 3
**Confidence:** 4

**Main Review:**

Review of "ONE FOR MANY: AN INSTAGRAM INSPIRED BLACKBOX ADVERSARIAL ATTACK"

This paper presents a genetic algorithm (GA)  based method to generate adversarial images, i.e., images perturbed in a way such that it leads to misclassifications in a number of image classification models.
Investigation of GAs in the context of adversarial image generation task is a novel and interesting one.
It's however difficult to see the connection of this with Instagram filters. Why is a sequence of filtering required (Eqn 2) via composition of filters?

Moreover, more details need to be included on the GA process. How is a particular state represented (each row indicating filter parameters)? Details need to be included on what happens during the cross-over and the mutation operations? Do cross-over operations exchange filter parameter values? Does all the filters use the same number of parameters?

The paper mentions about "gradient towards a better solution". However, GA-based approaches don't require computing gradients.

Another concern is the time it would take to generate one adversarial example. Evaluation of the fitness function of each state needs to be evaluated by making predictions
with the target models, which indicates that executing GA in this case is going to be time consuming.

From a novelty perspective, the paper is quite limited because algorithm is also based on the AGV paper (Baia et. al. 2021). The only difference is that multiple target networks are used. It's not clear why (Baia et. al. 2021) wasn't used as a baseline. Moreover, there is no rational explanation provided on why zooming in on an AGV generated image wouldn't produce the visible artefacts --- what characteristics of the proposed approach is likely to ensure this?


**Summary Of The Paper:**

This paper presents a genetic algorithm (GA)  based method to generate adversarial images, i.e., images perturbed in a way such that it leads to misclassifications in a number of image classification models.
Investigation of GAs in the context of adversarial image generation task is a novel and interesting one.



**Summary Of The Review:**

The strengths of the paper (extensive experiments) hasn't been able to outweigh the weaknesses (limited novelty, lack of AGV baseline, a seemingly slow approach).

---

> ### Author Response · Authors · 2021-11-16
> **Response to Reviewer chwU (part 1)**
>
> We thank the review for the useful feedback on our paper. We address the concerns below.
>
> **It's however difficult to see the connection of this with Instagram filters. Why is a sequence of filtering required (Eqn 2) via composition of filters?**
>
> We think that Instagram filters are very common and so frequently used to be considered an interesting case study.
> In previous papers (for ex. Destylization [1]) has been shown that Instagram inspired filters can have adversarial, but limited effects, when applied singularly. Hence, we decided to study the effects of filter composition with the aim to increase the adversarial effectiveness. Moreover, in the usual photo editing process people use more than 1 filter in order to obtain the desired effect. We just wanted to simulate a more realistic scenario. Our motivation is inspired by a future app able to apply enhancing filters (to an image) to have an  aesthetically pleasing effect able, at the same time, to protect user privacy by attacking information extraction applications based on NN.
>
> We added a sentence in section Introduction about this.
>
> [1] Recognizing Instagram Filtered Images with Feature De-stylization. 2020, https://arxiv.org/pdf/1912.13000.pdf
>
> **Moreover, more details need to be included on the GA process. How is a particular state represented (each row indicating filter parameters)**
> We did not provide details in the submitted paper in order to save space but, considering your comment, we realized that the population element structure needs more explanation . In the new version we added additional details in section 3.3 Algorithm.
>
> Given a set $S=\{f_1,f_2,···f_m\}$ of image filters, the outer algorithm *genotype* (with length $n$) is encoded as a list of $n$ integers $k_1,\ldots,k_n \in \{1,\ldots,m\}$  representing the corresponding filters in $S$, while the inner algorithm *genotype* is represented by a list containing the pairs of parameters used for each selected filter
> $((\alpha_1,s_1), \ldots, (\alpha_n,s_n))$. The associated *phenotype* is sequence of parametrized filters able to generate the adversarial examples by applying the selected sequence of filters, with their corresponding optimized parameters, as described in Eq.2.
>
> **Details need to be included on what happens during the cross-over and the mutation operations. Do cross-over operations exchange filter parameter values?**
>
> The implemented crossover is a standard one-point crossover. It creates a new element by taking a part from the first parent and a part from the second parent (included their parameters) according to the splitting random point. For example, given two elements $P_1=(f'_1(\alpha'_1,s'_1), \ldots, f'_n(\alpha'_n,s'_n))$ and $P_2=(f''_1(\alpha''_1,s''_1), \ldots, f''_n(\alpha''_n,s''_n))$ as parents and using a crossover index $i=2$, we obtain the child element $(f'_1(\alpha'_1,s'_1),f'_2(\alpha'_2,s'_2),f''_3(\alpha''_3,s''_3), \ldots, f''_n(\alpha''_n,s''_n))$.
>
> During the mutation phase, mutation can be applied to each filter (according to the mutation probability) that, in case, is substituted with another filter initialized with random parameter values. For example considering the element $P=(f_1(\alpha_1,s_1), f_2(\alpha_2,s_2), f_3(\alpha_3,s_3)\ldots, f_n(\alpha_n,s_n))$
> and supposing that filters $f_1$ and $f_3$ have been chosen to mutate with $g_1$ and $g_2$,
>  the new element is $P^*=(g_1(\alpha^*_1,s^*_1), f_2(\alpha_2,s_2), g_2(\alpha^*_2,s^*_2)\ldots, f_n(\alpha_n,s_n))$, where $\alpha^*_i,s^*_i$ are randomly extracted from the parameter domains.
>
>  In the new version of the paper Crossover and mutation descriptions in subsection 3.3.1 are extended with examples and additional details.
>
> **Does all the filters use the same number of parameters?**
>
>  Yes, all the filters use two parameters, *strength* and *intensity*, as described in section 2.2.
>
> **The paper mentions about "gradient towards a better solution". However, GA-based approaches don't require computing gradients.**
>
> We suppose that this comment refers to the sentence in section 3.3.2.
> In this case we refer to the optimization process of the ES strategy in the inner algorithm that, as in our implementation, can use a gradient-based approach to optimize the elements. So, we are speaking about the gradient of the internal fitness function and not the gradient of the attacked neural network.

---

> ### Author Response · Authors · 2021-11-16
> **Response to Reviewer chwU (part 2)**
>
> We thank the review for the useful feedback on our paper. We address the concerns below.
>
> **Another concern is the time it would take to generate one adversarial example. Evaluation of the fitness function of each state needs to be evaluated by making predictions with the target models, which indicates that executing GA in this case is going to be time consuming.**
>
> We were aware that the use of population based algorithms could have an impact from the computational point of view, so we posed particular attention to the number of queries (and consequently fitness evaluations) needed to craft effective attacks, during the phase of algorithm and population design.
> As you can note in Section 3.4.1 the average number of queries necessary to craft an effective attack is low, and it is lower than other Black-box approaches. The drawback of using an evolutionary approach is highly mitigated by the needed reduced number of generations and population size.
> In order to explain better this point we added, in the new version of the paper, the section 4.2 "Query Efficiency" where some comparisons with state of the art methods are reported.
>
> Just to give an idea about the time used: the version MultinetworkAGV-5Net-5filters  takes about 50s to make an external generation querying all the 5 victim models (160 queries for each model) with the experiment configuration reported in the paper. The experiments run on a GPU RTX3090.
>
> **From a novelty perspective, the paper is quite limited because algorithm is also based on the AGV paper (Baia et. al. 2021). The only difference is that multiple target networks are used. It's not clear why (Baia et. al. 2021) wasn't used as a baseline.**
>
> The algorithm introduced in (Baia et al 2021) proposes a universal approach able to find a single image agnostic perturbation for a whole set of images. Since our goal was to improve the transferability property
> we decided to follow the per-instance approach and to exploit the characteristic of the evolutionary algorithm of having a general form objective function (fitness function) to include more target networks.
> Said that, we can point out that the experimental results in the second column (Single Network) of Table 1 are basically obtained by the algorithm proposed by (Baia et al 2021) when applied to a single image. It can be considered our baseline. In order to clarify this, we added a sentence in Section 4.3 in the new version of the paper.
>
> **Moreover, there is no rational explanation provided on why zooming in on an AGV generated image wouldn't produce the visible artefacts --- what characteristics of the proposed approach is likely to ensure this?**
>
> The absence of artifacts in our filtered images derives from the uniform/ smooth nature of the filters. The attacks produced by algorithms like (Dong et al. 2019, Wu et al. 2020) add noise that is optimized just considering the total amount, or the maximum amount, of pixel modifications. These attacks show, in general, sharp differences between adjacent pixels: this can happen because close pixels with very similar (at most the same) values can be modified in very different way, breaking  color graduation and its smoothness.

---

### Author Response · Authors · 2021-11-17
**General Response to Reviewers**

We want to sincerely thank all the reviewers for the valuable feedback and comments that help to improve the quality of our work.

We have made the following changes to our article, highlighted in  red:

- clarified the motivation behind the idea of using a combination of filters, in section 1 Introduction
- added Figure 6 in Appendix that shows the effects produced by the ten image filters individually
- added more details regarding the population element structure, crossover and mutation operations, in section 3.3 Algorithm
- added the section 4.2 Query efficiency where we make comparisons with state-of-the-art methods
- added a clarification about the baseline method, in section 4.3 Attacks and Transferability
- updated the Table 3 which now reports also the minimum transferability values and the corresponding attacked network
- added section 4.4 Stealthiness of Adversarial Examples where we address and evaluate the stealthiness of adversarial examples
- added section 4.5 Deception Ability, where we discuss the strength of the attack as the ability to move the wrong classification away from the correct one.

Please refer to the individual responses for more details.

---

### Decision · Program_Chairs · 2022-01-20

**Decision:**

Reject

**Comment:**

This work proposed a nested evolutionary algorithm to choose image filters and filter parameters for back-box attacks, with the emphasize of high transferability.

After reading the manuscript, the comments of reviewers and the authors' responses, I think the main issues of this work include:
1. The limited novelty of the main idea, since there have been many filter-based attacks, and this work is very close to an existing work;
2. The solution is not new, since the evolutionary method is also well adopted in adversarial attacks;
3. Many many black-box attack methods are not cited and compared, though the authors argued that their perturbation upper bound are different such that they cannot be compared, which is not convincing;
4. The claimed high transferability is not well explained, maybe due to the model ensemble (as indicated by reviewer eN8o). Besides, many existing works that studied transferability are not cited and compared.
5. Experiments are inadequate. The authors added some results in the revised version, but the current shape is still not ready for publication.

Thus, my recommendation is reject. Hope the reviews can help to improve this work in future.